# Characterizing Periodic Variations of Atomic Frequency Standards via Their Frequency Stability Estimates

**DOI:** 10.3390/s23115356

**Published:** 2023-06-05

**Authors:** Weiwei Cheng, Guigen Nie, Jian Zhu

**Affiliations:** 1School of Transportation, Civil Engineering and Architecture, Foshan University, Foshan 528000, China; lucifercheng87@whu.edu.cn; 2GNSS Center, Wuhan University, Wuhan 430079, China; 3Collaborative Innovation Center for Geospatial Information Technology, Wuhan University, Wuhan 430079, China; 4Hubei Luojia Laboratory, Wuhan 430079, China

**Keywords:** GPS satellite clocks, periodic variations, frequency stability, Allan variance, Hadamard variance

## Abstract

The onboard atomic frequency standard (AFS) is a crucial element of Global Navigation Satellite System (GNSS) satellites. However, it is widely accepted that periodic variations can influence the onboard AFS. The presence of non-stationary random processes in AFS signals can lead to inaccurate separation of the periodic and stochastic components of satellite AFS clock data when using least squares and Fourier transform methods. In this paper, we characterize the periodic variations of AFS using Allan and Hadamard variances and demonstrate that the Allan and Hadamard variances of the periodics are independent of the variances of the stochastic component. The proposed model is tested against simulated and real clock data, revealing that our approach provides more precise characterization of periodic variations compared to the least squares method. Additionally, we observe that overfitting periodic variations can improve the precision of GPS clock bias prediction, as indicated by a comparison of fitting and prediction errors of satellite clock bias.

## 1. Introduction

The onboard atomic frequency standard (AFS), commonly referred to as the satellite clock, is a crucial component of the Global Navigation Satellite System (GNSS) satellites. This is due to the close relationship between their characteristics and the system’s geodetic performance [1,2,3]. While ground-based AFS processing methodologies such as the power-law noise model [3], the offset and drift model for atomic clocks [4], Allan variance (AVAR) [5], and Hadamard variance (HVAR) [6] are useful in dealing with onboard clock data, they may not be sufficient. The behavior of satellite AFS differs from that of ground AFS due to an integrated non-dispersive effect affecting the broadcast timing signals observed from the ground [2,5,7,8,9,10]. For instance, sub-daily periodic components can cause a clock bias up to 7.80±1.61 ns [5] (about 2.33±0.48 m equivalent light travel distance). Therefore, a more comprehensive modeling of periodic variations in GNSS satellite clocks is necessary to optimize the current system and plan future improvements.

GNSS satellite clocks are aligned with terrestrial time (TT) [11,12,13,14], which is annually realized by the Bureau International des Poids et Mesures (BIPM) as “TT BIPM∗∗” (“∗∗” indicates the year of calculation) [15]. Although TT is currently the most accurate and stable timescale, the post-processed timescale is not suitable for real-time applications. The International Atomic Time (TAI), established by the Consultative Committee for Time and Frequency (CCTF) and maintained by BIPM after the adoption of the atomic definition of second in the 13th General Conference on Weights and Measures (CGPM), is probably the most reliable “real-time” timescale in application. It is generated by more than 400 atomic clocks spread globally as an EAL (échelle atomique libre, or free atomic timescale), and the frequency of the EAL is steered by measurements of 12 primary or secondary frequency standards [15]. While GNSS timescales (e.g., GPS [11], GLONASS [12], Beidou [13], Galileo [14], etc.) are aligned with TAI in the long-term frequency, each satellite system has its own timescale. Therefore, the International GNSS Service (IGS) developed its own timescale, the International GNSS Service timescale (IGST), for GNSS research and application. IGST is also aligned with TAI in the long-term frequency [16]. Additionally, both the timescale algorithm of merely onboard AFS [4] and the IGST generated from both onboard and ground AFS [16] use the same time offset-frequency offset-drift (or aging) clock model as TAI [15,17]. Although time and frequency metrology techniques are essential to GNSS applications, it should be mentioned that the GNSS technique plays a significant role in resolving demands for universal timescales following scientific and industrial progress [18].

The behavior of space AFS is different from that of ground AFS [3]; however, electrical, gravitational, atmospheric, and thermal variations can cause magnetic field [19,20], microwave energy [21], light [22], thermally induced frequency shifts [23] of the clock. For instance, oblateness of geopotential causes periodic variations whose fundamental frequency is 2 cpd (cycles per day) in GPS satellite AFS [7]. Therefore, a satellite clock should be considered as the integration of AFS and “non-dispersive effects of all satellite components that affect the broadcast timing signals as observed from the ground” [5].

In fact, Senior et al. reported the detection of periodic signals at n×[2.0029±0.0005] cpd for n=1, 2, 3, and 4 [5] in all GPS Block II and Block IIA cesium and rubidium and Block IIR and IIR-M rubidium clocks. The variations were also reported in Block IIF rubidium clock and Block IIF cesium clock by Montenbruck et al. [9] and Fan et al. [2], respectively. While Senior et al. point out that 12-hour variation has a satellite-type-dependent relation with the sun–spacecraft–earth angle [5] and Montenbruck et al. suggest that solar illumination is the root cause of 12-hour period Block IIF Rb variations, they can not explain the “robust” [5] difference (58 ± 22 s) between the fundamental frequency (a weighted mean) of sub-daily GPS periodics (n×[2.0029±0.0005] cpd) and the GPS mean orbit period (2.0057 cpd) [24]. However, the reason for this discrepancy remains unexplained. In addition, these studies are based on Fourier transform and least squares methods:Satellite clock bias includes both deterministic and random signals: while the spectra of deterministic signals are estimated from its direct Fourier transform, the spectra of stochastic processes are calculated from the Karhunen–Loeve transform of its covariance [25]. On the other hand, Dong et al. show that the Karhunen–Loeve transform of covariance can distort the spectral response of deterministic signals [26].Additionally, Zhou et al. reported that there is no uniform method based on least squares for detecting, fitting, and removing periodic variations in Beidou satellite system (BDS) inclined geosynchronous orbit (IGSO), geostationary earth orbit (GEO), and medium earth orbit (MEO) satellite clock periodic variations [27].

Instead of the frequency-domain Fourier transform and least squares methods, we study the characterization of GPS onboard AFS periodic variations using frequency stability in this paper. In Section 2, we demonstrate that if the AFS signal comprises sinusoidal and random signals, the AVAR and HVAR of the sinusoidal signal are independent of those of the random signals. Additionally, AVAR and HVAR of GPS AFS periodic variations are formulated, and the method of fitting periodic variations from AVAR and HVAR estimates are presented. In Section 3, the effectiveness of the proposed model is tested against both simulated and real clock data.

## 2. Methods

To characterize the periodic variations of GNSS satellite AFS using its frequency stability estimates, we derive the AVAR and HVAR of periodic variations in this section. During the process, we demonstrate that the frequency stability of periodic variations is independent of the frequency stability of the stochastic component of AFS signals. Finally, we present the numerical method utilized in this paper for characterizing periodic variations of AFS through frequency stability estimates.

### 2.1. Allan and Hadamard Variances of Periodic Variations

Suppose the satellite AFS clock bias, x(t), is composed of stochastic and periodic signals, i.e.,
$$x\left(t\right)=\tilde{x}\left(t\right)+a\sin\left(\omega t+\varphi_{0}\right)=\tilde{x}\left(t\right)+\bar{x}\left(t\right),$$
then the theoretical values of AVAR and HVAR of the frequency standard are:(1)$$E\left[\sigma_{AVAR}^{2}\left(x,\tau \right)\right]=E\left\{\left.\frac{{\left[\Delta_{2}H\left(\bar{x}\left(t\right),\tau \right)\right]}^{2}-2\Delta_{2}H\left(\bar{x}\left(t\right),\tau \right)\Delta_{2}H\left(\tilde{x}\left(t\right),\tau \right)+{\left[\Delta_{2}H\left(\tilde{x}\left(t\right),\tau \right)\right]}^{2}}{2m^{2}\tau^{2}}\right|\tau ,\omega \right\}$$
and
(2)$$E\left[\sigma_{HVAR}^{2}\left(x,\tau \right)\right]=E\left\{\left.\frac{{\left[\Delta_{3}H\left(\bar{x}\left(t\right),\tau \right)\right]}^{2}-2\Delta_{3}H\left(\bar{x}\left(t\right),\tau \right)\Delta_{3}H\left(\tilde{x}\left(t\right),\tau \right)+{\left[\Delta_{3}H\left(\tilde{x}\left(t\right),\tau \right)\right]}^{2}}{6m^{2}\tau^{2}}\right|\tau ,\omega \right\}$$
respectively, where $\tilde{x}\left(t\right)$ is the random part of x(t), a superposition of power-law noise (PLN) processes. Next, $\bar{x}\left(t\right)$ denotes the periodic variations of the clock:$$\bar{x}\left(t\right)=a\sin\left(\omega t+\varphi_{0}\right).$$For the sake of brevity, we define two auxiliary functions, $\Delta_{2}H\left(x,\tau \right)$ and $\Delta_{3}H\left(x,\tau \right)$:$$\Delta_{2}H\left(x(t),\tau \right)=x\left(t+2\tau \right)-2x\left(t+\tau \right)+x\left(t\right),$$
$$\Delta_{3}H\left(x(t),\tau \right)=x\left(t+3\tau \right)-3x\left(t+2\tau \right)+3x\left(t+\tau \right)-x\left(t\right),$$
respectively. The averaging interval $\tau =m\tau_{0}$, *m* should be a positive integer, and $\tau_{0}$ is the sampling interval of the data.

By the theory of statistics, for a fixed averaging interval τ,
(3)$$E\left[\left.\Delta_{2}H\left(\bar{x}\left(t\right),\tau \right)\Delta_{2}H\left(\tilde{x}\left(t\right),\tau \right)\right|\tau \right]=E\left[\left.\Delta_{2}H\left(\bar{x}\left(t\right),\tau \right)\right|\tau \right]E\left[\left.\Delta_{2}H\left(\tilde{x}\left(t\right),\tau \right)\right|\tau \right].$$Given a PLN process, $f^{\alpha }$, where *f* denotes the Fourier frequency, $f^{\alpha }$ shapes the power spectral distribution (PSD) of the PLN process. When α>−3, $E\left[\left.\Delta_{2}H\left(\bar{x}\left(t\right),\tau \right)\right|\tau \right]=0$; that is:

**Theorem** **1.**
*For signals consistin of PLN processes, $f^{\alpha }$, α>−3, and sinusoidal variations, the AVAR of the periodic variations is independent of the AVAR of the random signals.*


Similarly, when α>−5, $E\left[\left.\Delta_{3}H\left(\bar{x}\left(t\right),\tau \right)\right|\tau \right]=0$. As a consequence:

**Theorem** **2.**
*For signals consisting of PLN noise processes, $f^{\alpha }$, α>−5, and sinusoidal variations, the HVAR of the periodic variations is independent of the HVAR of the random signals.*


By applying Theorem 1 to Equation (Equation 1), we can separate the AVAR of stochastic fluctuations from the one of sinusoidal variations:$$E\left[\sigma_{AVAR}^{2}\left(x,\tau \right)\right]=E\left[\sigma_{AVAR}^{2}\left(\tilde{x},\tau \right)\right]+E\left[\sigma_{AVAR}^{2}\left(\bar{x},\tau \right)\right].$$

Since any continuous periodic signals in $\ell^{2}$-space can be decomposed as the summation of a series of sinusoidal signals, AVAR of periodic variations of the satellite AFS is independent of those of the stochastic vibrations of the clock. There are plenty of documents on the characterization of AFS clock random vibrations; we will focus on characterizing periodic variations of AFS clock using AVAR.

After expanding $E\left[\sigma_{AVAR}^{2}\left(\bar{x},\tau \right)\right]$ with the definitions of $\bar{x}\left(t\right)$ and $\Delta_{2}H\left(x(t),\tau \right)$ and Equation (Equation 1), we have:$$\begin{array}{ccc} E\left[\sigma_{AVAR}^{2}\left(\bar{x},\tau \right)\right] & = & \frac{a^{2}}{2\tau^{2}}E\left[\sin^{2}\left(2\omega \tau +\omega t+\varphi_{0}\right)-4\sin\left(\omega \tau +\omega t+\varphi_{0}\right)\sin\left(\omega t+\varphi_{0}\right)\right.\\ & & +4\sin^{2}\left(\omega \tau +\omega t+\varphi_{0}\right)-4\sin\left(2\omega \tau +\omega t+\varphi_{0}\right)\sin\left(\omega \tau +\omega t+\varphi_{0}\right)\\ & & \left.\left.+\sin^{2}\left(\omega t+\varphi_{0}\right)+2\sin\left(2\omega \tau +\omega t+\varphi_{0}\right)\sin\left(\omega t+\varphi_{0}\right)\right|\tau ,\varphi_{0}\right] \end{array}$$

Since *t* goes from −∞ to +∞, the value of the initial phase, $\varphi_{0}$, makes no difference in the computation of $E\left[\sigma_{AVAR}^{2}\left(\bar{x},\tau \right)\right]$. By replacing $\omega t+\varphi_{0}$ with ωt and taking the following relations into account,
$$E\left[\sin(\omega \tau +\omega t)\sin(\omega t)\right]=E\left[\sin(2\omega \tau +\omega t)\sin(\omega \tau +\omega t)\right]=\frac{1}{2}\cos\left(\omega \tau \right),$$
$$E\left[\sin(2\omega \tau +\omega t)\sin(\omega t)\right]=\frac{1}{2}\cos\left(2\omega \tau \right),$$
and
$$E\left[\sin^{2}\left(2\omega \tau +\omega t\right)\right]=E\left[\sin^{2}\left(\omega \tau +\omega t\right)\right]=E\left[\sin^{2}\left(\omega t\right)\right]=\frac{1}{2},$$
we formulate the AVAR of periodic variations as following:(4)$$E\left[\sigma_{AVAR}^{2}\left(\bar{x},\tau \right)\right]=\frac{3-4\cos(\omega \tau )+\cos(2\omega \tau )}{2a^{-2}\tau^{2}}$$Similarly, the mathematical expectation of HVAR for the periodic variations can be formulated as following:(5)$$E\left[\sigma_{HVAR}^{2}\left(\bar{x},\tau \right)\right]=\frac{10-\cos(3\omega \tau )+6\cos(2\omega \tau )-15\cos(\omega \tau )}{6a^{-2}\tau^{2}}$$

Numerical computation of Equations (Equation 4) and (Equation 5) may result in negative values due to truncation errors. To prevent numerical evaluations of Equation (Equation 4) from being negative values, we recast Equation (Equation 4) as a summation of squares:$$\sigma_{x}^{2}\left(m\tau_{0}\right)=\frac{{\left[\cos(km)-2\cos(2km)+1\right]}^{2}+{\left[\sin(km)-2\sin(2km)\right]}^{2}}{4a^{-2}m^{2}\tau_{0}^{2}}$$

Similarly, the mathematical expectation of HVAR for periodic variations, Equation (Equation 5), can be reformulated as following:$$\begin{array}{cc} \sigma_{z}^{2}\left(m\tau_{0}\right)= & \frac{{\left[1-3\cos(km)+3\cos(2km)-\cos(3km)\right]}^{2}}{12a^{-2}m^{2}\tau_{0}^{2}}\\ & +\frac{{\left[3\sin(km)-3\sin(2km)+\sin(3km)\right]}^{2}}{12a^{-2}m^{2}\tau_{0}^{2}}. \end{array}$$

### 2.2. Characterizing Periodic Variations Using Frequency Stability Estimates

To evaluate periodic variations from frequency stability estimates of GNSS satellite AFS, we modify the stochastic model proposed in [28]:(6)$$\begin{array}{ccc} \min & \; & {\left(\Phi_{b}h+\Phi_{s}a-\sigma \right)}^{T}Q\left(\Phi_{b}h+\Phi_{s}a-\sigma \right),\\ s.t. & \; & \left[\begin{array}{c} B\left(\varepsilon \right)h+\Phi_{s}a-\sigma \\ -B\left(1-\varepsilon \right)h-\Phi_{s}a+\sigma \\ -a\\ -h\\ -a \end{array}\right]\leq 0. \end{array}$$
where “s.t.” is the abbreviation of “subject to”; σ is a column vector of frequency stability estimates (when the word “vector” is used, we mean column vector by default):$$\sigma ={\left[\sigma_{k_{1}}^{2}\left(\tau_{0}\right)\;\sigma_{k_{1}}^{2}\left(2\tau_{0}\right)\;\cdots \sigma_{k_{1}}^{2}\left(m_{1}\tau_{0}\right)\;\sigma_{k_{2}}^{2}\left(\tau_{0}\right)\;\sigma_{k_{2}}^{2}\left(2\tau_{0}\right)\;\cdots \sigma_{k_{2}}^{2}\left(m_{2}\tau_{0}\right)\;\cdots \right]}^{T}.$$Subscript *k* is used as a generic form that indicates different kinds of frequency stability:(7)$$\sigma_{k}^{2}\left(\tau \right)=\int_{0}^{\infty }S_{x}\left(f\right){\left|H_{k}\left(f\right)\right|}^{2}df=\sum_{i=1}^{N_{h}}\Phi_{k}\left(\alpha_{i},\tau \right)h_{\alpha_{i}},$$$H_{k}\left(f\right)$ is the transfer function of $\sigma_{k}^{2}\left(\tau \right)$, $S_{x}\left(f\right)$ is the PSD of PLN,
(8)$$S_{x}\left(f\right)=\sum_{i=1}^{N_{h}}h_{\alpha_{i}}f^{\alpha_{i}}=\left(2\pi f^{2}\right)S_{x}\left(f\right).$$The variable *h* is a vector of noise intensity coefficients, $h_{\alpha_{i}}$, $\alpha_{i}=2$, 1,⋯,−4, $i=1,\cdots ,\alpha_{N_{h}}$, corresponding to white PM (phase modulation), flicker PM, white frequency modulation (WHFM), flicker FM (FLFM), random walk FM (RWFM), and random run FM, respectively.
$$\Phi_{s}=\left[\begin{array}{cccc} E\left[\sigma_{k_{1}}^{2}\left(\sin\left(\omega_{1}t\right),\tau_{0}\right)\right] & E\left[\sigma_{k_{1}}^{2}\left(\sin\left(\omega_{2}t\right),\tau_{0}\right)\right] & \cdots & E\left[\sigma_{k_{1}}^{2}\left(\sin\left(\omega_{l}t\right),\tau_{0}\right)\right]\\ E\left[\sigma_{k_{1}}^{2}\left(\sin\left(\omega_{1}t\right),2\tau_{0}\right)\right] & E\left[\sigma_{k_{1}}^{2}\left(\sin\left(\omega_{2}t\right),2\tau_{0}\right)\right] & \cdots & E\left[\sigma_{k_{1}}^{2}\left(\sin\left(\omega_{l}t\right),2\tau_{0}\right)\right]\\ \vdots & \vdots & \ddots & \vdots \\ E\left[\sigma_{k_{1}}^{2}\left(\sin\left(\omega_{1}t\right),m_{1}\tau_{0}\right)\right] & E\left[\sigma_{k_{1}}^{2}\left(\sin\left(\omega_{2}t\right),m_{1}\tau_{0}\right)\right] & \cdots & E\left[\sigma_{k_{1}}^{2}\left(\sin\left(\omega_{l}t\right),m_{1}\tau_{0}\right)\right]\\ E\left[\sigma_{k_{2}}^{2}\left(\sin\left(\omega_{1}t\right),\tau_{0}\right)\right] & E\left[\sigma_{k_{2}}^{2}\left(\sin\left(\omega_{2}t\right),\tau_{0}\right)\right] & \cdots & E\left[\sigma_{k_{2}}^{2}\left(\sin\left(\omega_{l}t\right),\tau_{0}\right)\right]\\ \vdots & \vdots & \ddots & \vdots \end{array}\right];$$*a* is a vector of amplitudes of sinusoidal variations, and $\Phi_{b}$ is a matrix comprised of $\Phi_{k}\left(\alpha_{i},\tau \right)$:$$\Phi_{k}\left(\alpha ,\tau \right)=\int_{0}^{\infty }f^{\alpha }{\left|H_{k}\left(f\right)\right|}^{2}df.$$B(ε) and B(1−ε) are matrices composed of $B_{k}\left(\alpha ,\tau ,\varepsilon \right)$: $$B_{k}\left(\alpha ,\tau ,\varepsilon \right)=\frac{F^{-1}\left({\mathrm{EDF}}_{k}\left(\alpha ,\tau \right)/2,\varepsilon \right)\cdot \Phi_{k}\left(\alpha ,m\tau_{0}\right)}{{\mathrm{EDF}}_{k}\left(\alpha ,\tau \right)},$$${\mathrm{EDF}}_{k}\left(\alpha ,\tau \right)$ is the equivalence degree of freedom (EDF): $${\mathrm{EDF}}_{k}\left(\tau \right)=\frac{2E{\left[{\hat{\sigma }}_{k}^{2}\left(\tau \right)\right]}^{2}}{\mathrm{Var}\left[{\hat{\sigma }}_{k}^{2}\left(\tau \right)\right]},$$$F^{-1}\left(\cdot \right)$ is the inverse of cumulative distribution function: $$F\left(\sigma_{k}^{2}\left(\tau \right)\leq {\hat{\sigma }}_{k}^{2}\left(\tau \right)\right)=\int_{\sigma_{k}^{2}\left(\tau \right)}^{{\hat{\sigma }}_{k}^{2}\left(\tau \right)}\frac{u^{{\mathrm{EDF}}_{k}\left(\tau \right)/2}}{ue^{-u}\Gamma \left({\mathrm{EDF}}_{k}\left(\tau \right)/2\right)}du.$$The independent variables of Equation (Equation 6) are *h* and *a*. To solve the optimization problem, we set ε=0.0025 and $Q={\left[\mathrm{diag}\left(\sigma \right)\right]}^{-2}$.

Since ε is nonzero, the feasibility of Equation (Equation 6) depends on the values of frequency stability estimates. When Equation (Equation 6) is infeasible, which means the optimization problem has no solution, we use the following alternative model:(9)$$\begin{array}{cc} \min & {\;∥\mu ∥}_{1}+{∥\nu ∥}_{1},\\ s.t. & \;\left[\begin{array}{c} B\left(\varepsilon \right)h+\Phi_{s}a-\mathrm{diag}\left\{\sigma \right\}\mu -\sigma \\ -B\left(1-\varepsilon \right)h-\Phi_{s}a-\mathrm{diag}\left\{\sigma \right\}\nu +\sigma \\ -h\\ -a\\ -\mu \\ -\nu \\ \nu -1 \end{array}\right]\leq 0. \end{array}$$The independent variables of Equation (Equation 9) are μ, ν, *h*, and *a*; μ and ν are auxiliary variables defined such that nonzero components of μ and ν indicate violations of the inequalities $B\left(\varepsilon \right)h+\Phi_{s}a\leq \sigma$ and $B\left(1-\varepsilon \right)h+\Phi_{s}a\geq \sigma$, respectively. Details about the optimization models, Equations (Equation 6) and (Equation 9), and their numerical solutions can be found in references [28,29].

## 3. Results

In this section, we test the proposed model, Equations (Equation 4) and (Equation 5), with both simulated and GPS clock data using the method described by Equations (Equation 6) and (Equation 9).

### 3.1. Simulated Data

To verify Equations (Equation 4) and (Equation 5), we generate sinusoidal signals with frequencies, i×2.0029, i=1, 2, 3 cpa, and list their amplitudes in the second column of Table 1. (The frequencies and amplitudes are set according to reference [5]). As demonstrated in Figure 1a–c, the AVAR computed from Equation (Equation 4) (represented by red dots in Figure 1) aligns with the one estimated from the simulated sinusoidal signals (represented by black solid lines in Figure 1). The same holds true for Hadamard variances (HVAR), though these results are not presented here in the interest of brevity.

In Figure 1d, AVARs estimated from simulated clock data and their “theoretical” values are represented as a black solid line and red dots, respectively. The ”theoretical” AVAR is computed as the sum of Equations (Equation 7) and (Equation 4), and the simulated satellite clock biases are comprised of white PM, flicker PM, white WHFM, FLFM, RWFM, and sinusoidal variations of 12-, 6-, and 4-hour cycles (whose AVARs are shown in Figure 1a–c, respectively). As shown in Figure 1d, the two AVARs are close to each other, except for the values around averaging times $10^{4}$ s. It can be observed from Figure 1a–c that the AVAR of 12-, 6-, and 4-hour sinusoidal variation reaches its maximum near averaging times $5\times 10^{3}$, $10^{4}$, and $2\times 10^{4}$ s, respectively. Discrepancies between the two AVARs can be explained by the interactions between stochastic and periodic variations due to the finite data set.

Furthermore, to compare the effectiveness of detecting periodic variations from satellite clock biases directly using the least squares method and from Equations (Equation 4) and (Equation 5) using Equations (Equation 6) and (Equation 9), the periodic variations in the simulated data are fitted and removed using both methods. The amplitudes given by the least squares method and Equations (Equation 6) and (Equation 9) are listed in the third and fourth columns of Table 1, and AVARs of the resulting periodic variation-free clock biases are plotted as a red dash–dot line and blue dots, respectively, in Figure 2. In the least squares estimation process of this article, the satellite clock biases are are firstly detrended by fitting and removing a second-order polynomial. Then, a 12-hour sinusoidal signal was fitted and removed, followed by a 6-hour sinusoidal signal, a 4-hour frequency signal, and, finally, a 3-hour sinusoidal signal. While the least squares method gives greater periodic variation amplitudes than the set values of the simulated variation amplitudes and the ones estimated by Equations (Equation 6) and (Equation 9), AVARs estimated from simulated satellite clock bias with periodics fitted and removed by the least squares method does not seem to be periodic-variation-free. It can be observed in Figure 2 that in comparison to the AVARs estimated from the simulated data (plotted as a black solid line in Figure 2), the “lump” near averaging intervals $5\times 10^{3}$ suggests an extra 4-hour variation. In other words, the least squares method overfits the periodic variations. On the other hand, AVARs computed from simulated data with periodic variations fitted and removed using Equations (Equation 6) and (Equation 9) are the most periodic-variation-free AVARs among the three. To have a closer look at the overfitting problem, the root mean square (RMS) of the detrended simulated clock bias, detrended simulated data with periodic variations fitted and removed using the least squares method, and detrended simulated data with periodic variations fitted and removed using Equations (Equation 6) and (Equation 9) are computed: the result are $8.02\times 10^{-9}$, $7.80\times 10^{-9}$, and $7.95\times 10^{-9}$ ns, respectively. While the least squares method provides the lowest fitting errors, it seems to overfit flicker noise with a 4-hour periodic variation.

### 3.2. GPS SVN63 Clock Data

To verify the effectiveness of Equations (Equation 4) and (Equation 5), we computed the AVAR of the GPS PRN01 (SVN63) onboard rubidium AFS, which is shown as a black solid line in Figure 3 and Figure 4. The IGS final combined precise clock and orbit data from MJD 56739.0 to MJD 56745.9965 [30,31] are used in the computation. In this paper, the preprocessing of all IGS satellite clock data includes:Removing the J2 relativistic effect using Kouba’s method [7] and cubic spline interpolation of IGS final combined orbit data;Removing the day boundary discontinuities using Yao et al.’s methods [32].

For comparison, AVARs of 12-, 6-, and 4-hour periodic variations are plotted as a purple dash–dot line, blue dots, and a red dash line, respectively, in Figure 3. The values of these AVARs are normalized to be tangent to the AVAR of SVN63 satellite AFS at the maxima of their AVARs.

It can be observed from Figure 3 that the local maxima and minima of AVARs estimated from SVN63 clock bias and the 12-hour sinusoidal coincide for averaging time $\tau \geq 10^{4}$ s, suggesting the influence of 12- and 6-hour periodic variations. This consistency reinforces the conclusion that GPS onboard atomic frequency standards are influenced by 12-hour periodic variations. To quantify the influence of periodic variations, the variations are fitted and removed using both the least squares method and Equations (Equation 6) and (Equation 9); AVARs computed from the resulting clock biases are represented as a red dash–dot line and blue dots, respectively, in Figure 4. It seems that the least squares method may overfit the 12-, 6-, 4-, and 3-hour periodic variations of the GPS Block IIR rubidium AFS with frequency noise: gaps between the black solid line (AVAR estimated from IGS final combined SVN63 clock data) and the red dash–dot line (AVAR resulting from the least squares processing) keep increasing after averaging time $2\times 10^{4}$. On the other hand, after processing with Equations (Equation 6) and (Equation 9), the sigma–tau plot of the resulting AVAR estimates is the closest to the classical sigma–tau plot of the AVAR calculated from the summation of white PM, flicker PM, white WHFM, FLFM, and RWFM noise processes.

### 3.3. Other GPS Clock Data

To verify the general applicability of Equations (Equation 4) and (Equation 5), we compute the AVARs of GPS PRN02∼32 onboard AFS using IGS final combined precise clock data, clock biases with periodic variations removed by the least squares method, and Equation (Equation 9); the resulting AVARs are drawn as a black solid line, a red dash–dot line, and blue dots, respectively, in Figure 5 and Figure 6. During the time-span of IGS clock data used in this study (from 23 March 2014 to 27 December 2020), several satellites, such as PRN04, PRN14, PRN18, PRN23, and PRN32, were replaced. In addition, some PRN satellites, such as PRN04, have been substituted several times. For brevity, we only chose the oldest and most recent SVN satellites of these satellites from the dataset.

By comparison with Figure 1, we can make the following observations from Figure 5 and Figure 6:For more than half of the satellites, the sigma–tau plot of the AVAR computed from clock bias with periodic variations removed using Equation (Equation 9) is closer to the standard sigma–tau plot of AVARs than the least squares method.“Lumps” of AVARs estimated from PRN01, PRN02, PRN05, PRN07, PRN11, PRN12, PRN13, PRN14 (SVN41), PRN15, PRN16, PRN17, PRN18 (SVN54), PRN20, PRN21, PRN22, PRN23 (SVN76), PRN25, PRN28, PRN30, PRN31, PRN32 (SVN23), and PRN32 (SVN70) clock bias with periodic variations removed using Equation (Equation 9) at an averaging time around $4\times 10^{4}$ s suggest underestimation of 24-hour periodic variations.“Lumps” of AVARs estimated from PRN01, PRN02, PRN03, PRN04 (SVN34), PRN04 (SVN74), PRN05, PRN06, PRN09, PRN12, PRN14 (SVN77), PRN18 (SVN75), PRN19, PRN23 (SVN60), PRN23 (SVN76), PRN24, PRN26, PRN27, PRN30, and PRN32 (SVN23) clock bias with periodic variations removed using Equation (Equation 9) at an averaging time around $2\times 10^{4}$ s suggest underestimation of 12-hour periodic variations.“Lumps” of AVARs estimated from PRN04 (SVN34), PRN04 (SVN74), PRN05, PRN07, PRN10, PRN11, PRN12, PRN14 (SVN41), PRN17, PRN18 (SVN75), PRN20, PRN21, PRN22, PRN24, PRN25, PRN27, PRN29, and PRN32 (SVN70) clock bias with periodic variations removed using Equation (Equation 9) at an averaging time around $10^{4}$ s suggest underestimation of 6-hour periodic variations.“Lumps” of AVARs estimated from PRN02, PRN04 (SVN34), PRN05, PRN06, PRN07, PRN08, PRN09, PRN11, PRN13, PRN14 (SVN41), PRN15, PRN16, PRN18 (SVN75), PRN19, PRN22, PRN23 (SVN60), PRN25, PRN32 (SVN23), and PRN32 (SVN70) clock bias with periodic variations removed using Equation (Equation 9) at an averaging time around $5\times 10^{3}$ s suggest underestimation of 4-hour periodic variations.“Lumps” of AVARs estimated from PRN09, PRN11, PRN13, PRN18 (SVN54), PRN18 (SVN75), PRN22, PRN24, PRN27, PRN29, PRN31, PRN32 (SVN23), and PRN32 (SVN70) clock bias with periodic variations removed using Equation (Equation 9) at an averaging time around $3\times 10^{3}$ suggest underestimation of 3-hour periodic variations.

In other words, the optimization model Equation (Equation 9) is “conservative”: it tends to underestimate periodic variations of GPS satellite AFS. In effect, a large portion of the numerical solutions of 4- and 3-hour variations returned by solving Equation (Equation 9) are zero. According to the theory of statistics, the least squares method makes huge assumptions about data structure and produces stable results. On the other hand, Equation (Equation 9) only requires the least violation of the 95% confidence intervals by the AVAR estimates; it does not require optimal estimation of the amplitudes for periodic variations. By comparison with AVARs computed using the least squares method, as shown in Figure 5 and Figure 6, Equation (Equation 9) underestimates the periodic variations of PRN01, PRN03, PRN04 (SVN34), PRN04 (SVN74), PRN08, PRN09, PRN10, PRN19 (SVN75), PRN20, PRN21, PRN26, PRN27, and PRN29.

It can also be observed from Figure 4, Figure 5 and Figure 6 that the AVARs computed from clock biases with periodic variations fitted and removed using Equation (Equation 9) are always smaller than the AVARs computed from IGS final combined clock biases, and the two coincide at averaging intervals greater than $4\times 10^{4}$ s. For most cases, this also holds for AVARs estimated from clock bias with periodic variations fitted and removed using the least squares method. There are only two exceptions:The gap between the AVAR estimated from PRN01 clock bias with periodic variations fitted and removed using the least squares method and the AVAR computed from PRN01 clock bias enlarges with increasing averaging interval. Since the tail of AVAR estimated from PRN01 clock bias with periodic variations removed using the least squares method has a similar shape to the AVARs of 12-hour sinusoidal variations, and the AVAR computed from PRN01 clock bias with periodic variations fitted and removed using Equation (Equation 9) suggests strong frequency noise, the discrepancies between the AVAR estimated from PRN01 clock bias with periodic variations fitted and removed using the least squares method and the AVAR computed from PRN01 clock bias is caused by overfitting the periodic variations by taking a portion of frequency noises as 12-hour variation.The AVAR estimated from PRN22 clock bias with periodic variations fitted and removed using least squares method is greater than the AVAR computed from IGS final combined PRN22 clock bias around averaging time $2\times 10^{4}$ s. Since AVARs computed from the three PRN22 clock biases increase with the averaging interval for $\tau \geq 2\times 10^{4}$ s, PRN22 AFS is influenced by strong FM noise processes. It seems that the least squares method overfits the periodic variations of PRN22 by taking a portion of frequency noises as 12-hour variation.

Therefore, Theorems 1 and 2 hold for GPS onboard AFS.

Overfitting of periodic variations can sometimes help improve GNSS onboard clock prediction, as shown in Table 2. By taking frequency noises as sinusoidals, the RMS of one-day GPS onboard atomic clock prediction under two-day observation is decreased. However, removing periodic variations based on Equations (Equation 6) and (Equation 9) can increase the prediction RMS. This could be due to several reasons:Overfitting of periodic variations can reduce the clock residuals caused by power–law noise processes. When periodic variations are removed, the interaction between random clock behaviors and periodic variations is suppressed, leading to an increase in clock residuals and prediction RMS.Only high-variability estimates (HVAR) are used in solving Equations (Equation 6) and (Equation 9), which may not capture all the periodic variations present in the data. The maximum averaging time of HVARs estimated from two-day GPS clock bias is $1.44\times 10^{4}$ s, while the first local minimum of Equation (Equation 5) appears at averaging interval $\tau =2\times 10^{4}$ s. This means that some periodic variations may not be captured by HVAR estimates and could contribute to an increase in the RMS prediction when removed.

## 4. Conclusions and Discussion

The detection of periodic variations in GNSS satellite AFS presents a dilemma: while the spectra of deterministic sinusoidals can be estimated directly from the Fourier transform, those of power–law noise processes, which are random, must be calculated from the Karhunen–Loeve transforms of their covariance. Currently, time domain methods are generally used in the evaluation of AFS and timescales. In this paper, we characterize the periodic variations of GNSS satellite AFS clock bias using AVAR and HVAR. In addition, we demonstrate that the AVAR and HVAR of sinusoidal signals present in the clock bias are independent of the AVAR and HVAR of the random vibrations of the satellite AFS. The method to detect periodic variations using these characterizations is given, and it is test against both simulated and real data.

The proposed method characterizes the periodic variations of satellite AFS more accurately than the least squares method, as demonstrated in both simulated and real clock data tests. Therefore, the method developed in this article can serve as a criterion for detecting periodic variations. The least squares method tends to overfit the 12-, 6-, 4-, and 3-hour frequency variations, resulting in sinusoidals that may have no physical meaning. However, the overfitting demonstrated by the least squares method suggests that by treating some of the frequency noise as periodic variations, we can improve GPS prediction precision in practice.

## Figures and Tables

**Figure 1 sensors-23-05356-f001:**
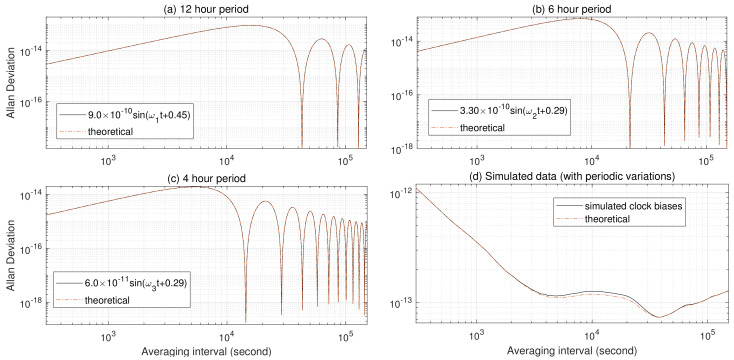
Allan variances of sinusoidal signals and simulated clock bias with periodic variations.

**Figure 2 sensors-23-05356-f002:**
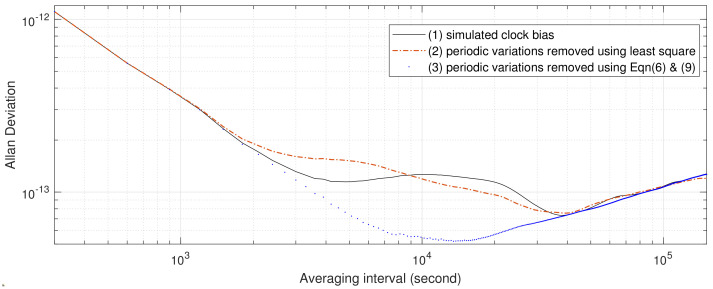
AVARs of simulated clock data 12-, 6-, 4-, and 3-hour periodic variations fitted and removed by least squares and by Equation (Equation 9), respectively.

**Figure 3 sensors-23-05356-f003:**
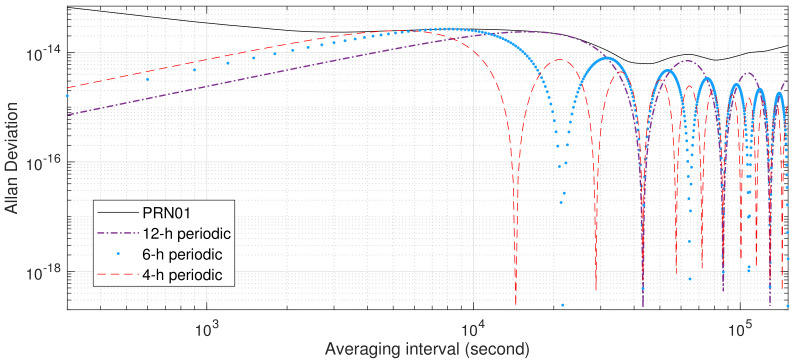
AVARs of GPS SVN63 onboard rubidium clock bias from MJD 56,739.0 to MJD 56,745.9965 12-, 6-, 4-, and 3-hour sinusoidal signals. AVARs of 12-, 6-, 4-, and 3-hour sinusoidal signals are magnified to be tangent to AVARs estimated from SVN63 clock bias at their maxima.

**Figure 4 sensors-23-05356-f004:**
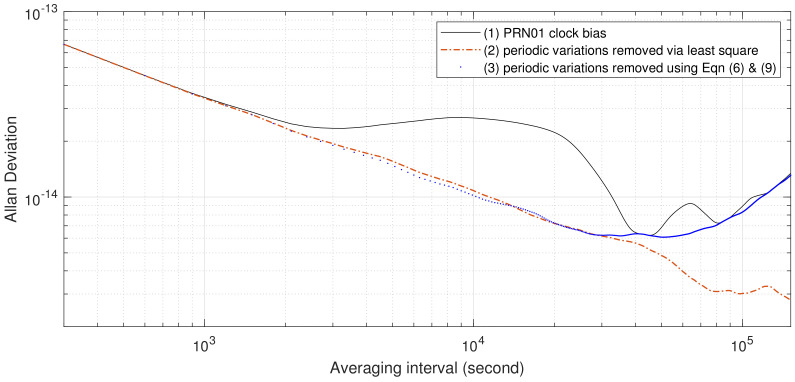
AVARs of GPS SVN63 onboard rubidium clock bias from MJD 56,739.0 to MJD 56,745.9965 12-, 6-, 4-, and 3-hour sinusoidals fitted and removed by least squares and by Equations (Equation 6) and (Equation 9), respectively.

**Figure 5 sensors-23-05356-f005:**
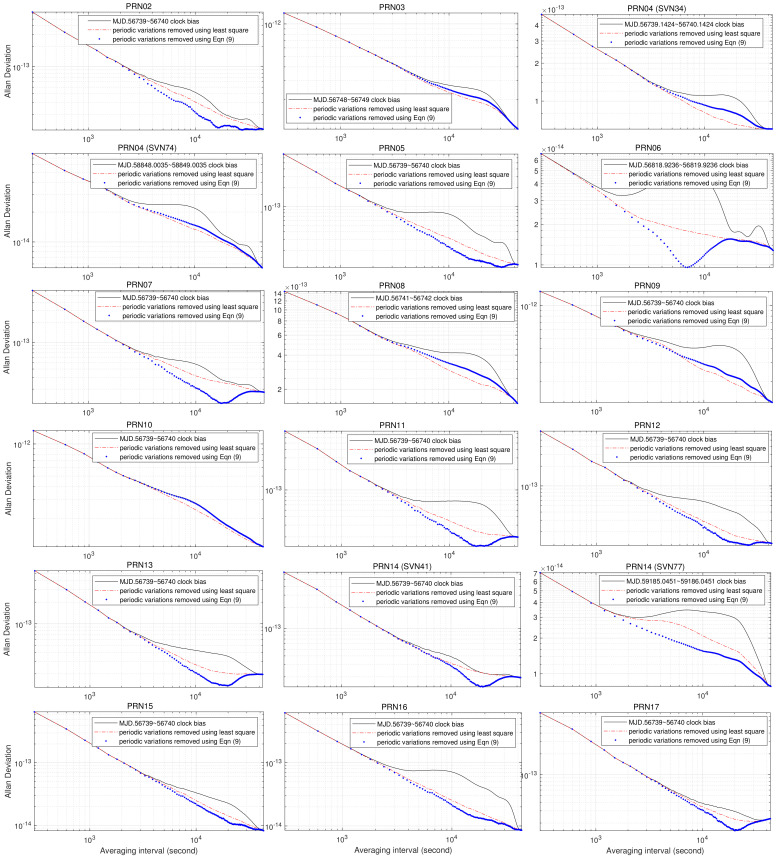
AVARs of GPS PRN02∼17 onboard clock bias 12-, 6-, 4-, and 3-hour sinusoidals fitted and removed using least squares method and Equation (Equation 9), respectively.

**Figure 6 sensors-23-05356-f006:**
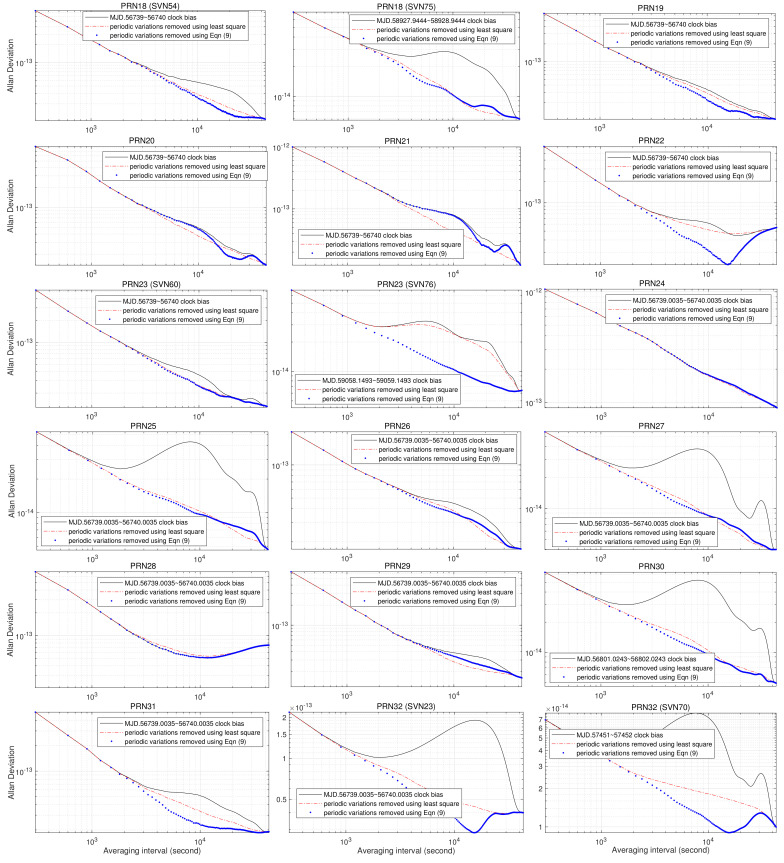
AVARs of GPS PRN18∼32 onboard clock bias 12-, 6-, 4-, and 3-hour sinusoidals fitted and removed using least squares method and Equation (Equation 9), respectively.

**Table 1 sensors-23-05356-t001:** Amplitudes of simulated periodic variations estimated by least squares method and solving Equation (Equation 6).

Frequency (cpa)	Input	Equations (Equation 6) and (Equation 9)	Least Squares
2.0029	$9.00\times 10^{-10}$	$8.92\times 10^{-10}$	$1.62\times 10^{-9}$
2×2.0029	$3.30\times 10^{-10}$	$3.40\times 10^{-10}$	$3.83\times 10^{-10}$
3×2.0029	$6.00\times 10^{-11}$	$1.35\times 10^{-10}$	$3.09\times 10^{-10}$
4×2.0029	0	0	$3.53\times 10^{-10}$

**Table 2 sensors-23-05356-t002:** One-day GPS clock bias prediction RMS (root mean square) of fitting and removing first- or second-order polynomial 12-, 6-, 4-, and 3-hour sinusoidals against two-day observations by using least squares and solving Equations (Equation 6) and (Equation 9), respectively.

PRN	With Periodics	Equations (Equation 6) and (Equation 9)	Least Square	Time-Span
G01	0.52	0.65	0.42	03-23-14∼12-27-20
G02	0.72	1.64	0.72	03-23-14∼12-27-20
G03	0.82	0.94	0.78	03-23-14∼12-27-20
G04	1.61	1.61	1.58	03-23-14∼12-27-20
G05	0.71	0.80	0.62	03-23-14∼12-27-20
G06	0.53	0.64	0.50	03-23-14∼12-27-20
G07	1.13	1.93	1.11	03-23-14∼12-23-20
G08	3.39	3.61	3.36	03-23-14∼12-27-20
G09	0.66	0.77	0.61	03-23-14∼12-27-20
G10	1.39	1.48	1.38	03-23-14∼12-27-20
G11	1.34	1.85	1.31	03-23-14∼12-27-20
G12	0.62	5.73	0.52	03-23-14∼12-27-20
G13	1.07	1.24	1.05	03-23-14∼12-27-20
G14	0.73	1.00	0.73	03-23-14∼12-25-20
G15	0.47	0.52	0.42	03-23-14∼12-27-20
G16	0.62	0.62	0.49	03-23-14∼12-27-20
G17	1.56	1.70	1.53	03-23-14∼12-27-20
G18	0.92	1.86	0.88	03-23-14∼12-27-20
G19	0.58	0.65	0.57	03-23-14∼12-27-20
G20	0.62	0.95	0.61	03-23-14∼12-27-20
G21	0.76	1.42	0.72	03-23-14∼12-26-20
G22	1.09	1.64	1.06	03-23-14∼12-27-20
G23	0.53	1.04	0.53	03-23-14∼12-27-20
G24	4.18	4.57	4.24	03-23-14∼12-27-20
G25	0.42	0.42	0.36	03-23-14∼12-26-20
G26	0.63	0.76	0.59	03-23-14∼12-27-20
G27	0.48	0.59	0.42	03-23-14∼12-27-20
G28	3.71	3.73	3.71	03-23-14∼12-27-20
G29	1.26	1.47	1.22	03-23-14∼12-27-20
G30	0.56	0.67	0.51	03-23-14∼12-27-20
G31	1.15	1.81	1.14	03-23-14∼12-27-20
G32	0.87	0.94	0.77	03-23-14∼12-27-20

## Data Availability

GNSS clock data are provided by the Multi-GNSS Experiment (MGEX) from the International GNSS Service (IGS) via http://garner.ucsd.edu/ (accessed on 27 October 2021).

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
