# Peer review of "Characterizing Periodic Variations of Atomic Frequency Standards via Their Frequency Stability Estimates"

_sensors, 2023, doi:10.3390/s23115356_

Round 1

Reviewer 1 Report

The authors characterize the periodic variations of atomic frequency via their frequency stability estimates.

(1)The proposed method is not clear, what is the developed model in the abstract?

(2) Allan and Hadamard are applied in AFS is not new. what is your contribution?

(3) The font sizes are not consistent in this paper, please check all

(4) When referencing equations, the "eq." or "equation" or only number is used,  please check all

(5) In figure 4, it is suggested to use color to descibe instead of "---" or "...."

Author Response

(1)The proposed method is not clear, what is the developed model in the abstract?

Response: In this paper, we formulate the AVAR and HVAR of periodic variations, and the proposed method refered to the method to extract the periodic variations of GNSS onboard clock from their AVAR and HVAR estimates. To clarify the question commented by the reviewer, we rewrite the abstract.

(2) Allan and Hadamard are applied in AFS is not new. what is your contribution?

Response: Allan and Hadamard are applied in AFS for characterizing their random vibrations. Our contribution is to apply AVAR and HVAR in characterization of periodic variations.

(3) The font sizes are not consistent in this paper, please check all

Response: We download the most recent LaTeX template offered by MDPI, and recomplied the article.

(4) When referencing equations, the "eq." or "equation" or only number is used,  please check all

All "eq." and "equation" are replaced by "Eqn".

(5) In figure 4, it is suggested to use color to descibe instead of "---" or "...."

Response: According to the suggestion of Reviewer 1,   "---" , "....", and "- . -" is replaced by "black solid line", "blue dots", and "red dash-dot line", respectively.

Reviewer 2 Report

This manuscript studies the timing and clock problem in GNSS satellites by conducting frequency stability estimation. The authors characterize the periodic variations of AFS and show that the Allan and Hadamard variances of the periodics are independent from the variances of the stochastic part. In the real experiment part, however, there is a lack of data to support the conclusion. Only 2 GPS PRNs with a fairly limit amount of time windows are included. I encourage the authors to conduct more experiments to backup your theory.

1. What is the main question addressed by the research?
The main question this work addressed is the AFS variation characterization and estimation

2. Do you consider the topic original or relevant in the field?

Yes

Does it address a specific gap in the field?

Yes

3. What does it add to the subject area compared with other published
material?

The authors try to address the problem using existing methods (priori work). “We test the methods they used in detecting and fitting periodic variations based in time domain in this paper. So the novelty of this paper may be questionable from the perspective of methodology.

4. What specific improvements should the authors consider regarding the
methodology? What further controls should be considered?

I had provided my improvements suggestion in my report. Other than emphasizing the novelty compared to prior work, I encourage the authors to conduct more experiments (different PRNs)

5. Are the conclusions consistent with the evidence and arguments presented
and do they address the main question posed?

No as there is lack of experiment accumulation to show the result is not cherry picking.

6. Are the references appropriate?

Yes

7. Please include any additional comments on the tables and figures.

The start time column in Table 2 can be consolidated into one value.

Extensive editing of English language are required as many gramma mistaken are seen across the manuscript. 

Author Response

1. Other than emphasizing the novelty compared to prior work, I encourage the authors to conduct more experiments (different PRNs)

Response: According to the suggestion of Reviewer 2, we conduct experiments with all PRNs. Since several satellites such as PRN04, PRN14, PRN18, PRN23, and PRN32 were replaced. Additionally, some satellites like PRN04 underwent rather high-paced replacements. Therefore, we chose the oldest and most recent clock bias values for these satellites from the dataset we used (from March 23, 2014 to December 27, 2020). 

2. The start time column in Table 2 can be consolidated into one value.

The start time column in Table 2 is consolidated into one value in the revised manuscript.

Reviewer 3 Report

Extensive editing of English language are required.

Author Response

  1.  All the abbreviations of ‘atomic frequency standard’ in the manuscript are different, such as ‘AFSs’ in line 1 and ‘AFS’ in line 13, which need to be consistent. Moreover, the abbreviations of ‘Allan variance’ and ‘Hadamard variance’ appears twice in lines 17 and 121, lines 18 and 123, please check all the other abbreviations.
    Response: ‘AFSs’ and ‘AFS’ are unified as 'AFS', the repeated appeareances of ‘Allan variance’ and ‘Hadamard variance’ are deleted.
    2.    All the abbreviations format should be standardized such as ‘TAI (International Atomic time)’ in line 28 should be changed to ‘International Atomic time (TAI)’. 
    Response: The abbreviations format is standardized as suggested.
    3.    The symbol ‘ ’ in Equation (6) or (9) looks very strange, which should be unified with the line 106 and 170. 
    Response: The vector inequality symbols in Equation (6) and (9) are replaced by the inequality symbols as suggested.
    4.    The title of Table 1 should be more specific and have an academic style. 
    Response: The title of Table 1 is rewritten.
    5.    The legends for figures 2 and 4 should be concise, and what the symbol ‘?’ means in the legend.
    Response: The question marks in the legends are corrected.
    6.    The word ‘periodics’ in line 5 should be ‘periodic variations’.
    Response: ‘The word periodics’ in Abstraction is replaced by ‘periodic variations’.
    7.    In this paper, the periodic variations of AFS are characterized, which show that the Allan and Hadamard variances of the periodic variations are independent from the variances of the stochastic part, but the whole work only displayed the Allan variances, how about the Hadamard variances?
    Response: For simulated data, Hadamard variances of the periodic variations is also independent from the variances of the stochastic part. 
    Since the sigma-tau plots of HVAR (of sinusoidal signals and simulated onboard clock data) are similar to those of AVAR, we choose not to present the plots.
    In the real data test, the use of HVAR is often lead to an underestimation of the variations.
    On the other hand, if we firstly removed periodic variations based on AVAR estimates, HVAR of clock bias and clock bias with periodic variations removed are consistent with the proposed formulation.
    8.    The whole works are mainly focus on only the PRN01 which can not fully support the conclusions, the more convincing work is to display the result of any other satellites.
    Response: According to the suggestion of Reviewer 2, we conduct experiments with all PRNs. Since several satellites such as PRN04, PRN14, PRN18, PRN23, and PRN32 were replaced. Additionally, some satellites like PRN04 underwent rather high-paced replacements. Therefore, we chose the oldest and most recent clock bias values for these satellites from the dataset we used (from March 23, 2014 to December 27, 2020).

Round 2

Reviewer 2 Report

I appreciate all authors for your effort to address my comments. With more data and experiment conducted for multiple GPS PRNs, the study became more solid with additional observation and conclusions. These new info are helpful for the community. Before proceeding to the publication, I encourage the authors to further work on the English writing of the manuscript. Some gramma mistakes are seen throughout the paper. Thank you! 

The authors is encouraged to further work on the English writing of the manuscript. Some gramma mistakes are seen throughout the paper. 

Author Response

We have thoroughly reviewed the article and made corrections to every grammar error, as well as improved the English expression in the revised version. However, we regret that we were unable to draw attention to the modifications made at the end of the paper due to time constraints.

Reviewer 3 Report

Thanks for taking care of the comments thoroughly. I have no more comments at this moment.

English should be further improved.

Author Response

(The authors gave the same response as above.)
